# Beyond Single Embedding: Modeling User Preferences as Distribution in Federated Recommendation

Chunxu Zhang [* 1 2 3]   Weipeng Zhang [* 1 2]   Guodong Long [4]   Zhiheng Xue [1 2]   Bo Yang [1 2]

## Abstract

Most federated recommender systems represent each user with a single embedding learned from local interaction data, implicitly assuming that user preferences are fixed and precisely identifiable. In federated settings, however, each client observes only a limited and fragmentary view of user behavior, rendering such point estimates inherently brittle. To address this mismatch, we model user preferences as distributions rather than points, allowing multiple compatible preference representations to coexist. Rather than collapsing evidence into a single embedding, our approach preserves uncertainty and diversity in user representations, providing a richer basis for preference modeling. We instantiate this idea with a diffusion-based generative framework that produces diverse user embeddings and derives recommendation scores by aggregating predictions across them. This distributional formulation yields more stable ranking behavior and improved robustness under ambiguous feedback. Extensive experiments on federated recommendation benchmark datasets demonstrate consistent and significant improvements over baselines. Our code is available at https://github.com/Zhangwp2420/FedDistRec.

## 1. Introduction

Federated Recommender Systems (FRS) coordinate model training across clients without sharing raw user inter-

---

[*]Equal contribution [1]College of Computer Science and Technology, Jilin University, Changchun, China [2]Key Laboratory of Symbolic Computation and Knowledge Engineering of Ministry of Education, Jilin University, Changchun, China [3]PolyU Academy for Artificial Intelligence, Hong Kong Polytechnic University, Hong Kong, China [4]Australian Artificial Intelligence Institute, FEIT, University of Technology Sydney, Sydney, Australia. Correspondence to: Bo Yang <ybo@jlu.edu.cn>.

*Proceedings of the $43^{rd}$ International Conference on Machine Learning*, Seoul, South Korea. PMLR 306, 2026. Copyright 2026 by the author(s).

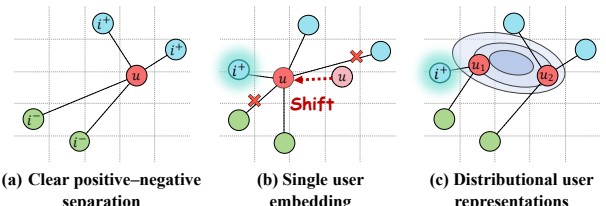

**(a) Clear positive–negative separation**   **(b) Single user embedding**   **(c) Distributional user representations**

*Figure 1.* Motivation behind distributional user representations. Each subfigure illustrates the geometric relationships between user representations and positive/negative items in the embedding space. **Left**: an ideal case with clear separation, where the user representation is close to positives and far from negatives. **Middle**: a single embedding shifts to fit an additional positive interaction, weakening its separation from earlier positives and moving closer to negatives. **Right**: distributional user representations use multiple embeddings to jointly satisfy conflicting constraints, maintaining robust positive–negative separation.

actions (Wang et al.; Long, 2024; Chen et al., 2025), thereby enabling privacy-preserving recommendation learning (Yang et al., 2020; Huang et al., 2021; Li et al., 2026a; Zhang et al., 2026). Most existing frameworks (Wang et al., 2022; Zhang et al., 2024; 2025a) rely on global parameters to capture population-level information and learn a single user embedding from local data to represent individual preferences. This formulation entails a strong determinacy assumption that a user's preference can be uniquely specified from a limited set of observed interactions. However, such interaction data are typically sparse and inherently ambiguous, making it difficult to justify committing to a unique and fixed preference representation.

As illustrated in Figure 1(a), recommendation models typically learn a single user embedding that is expected to separate observed positive items from negative ones in the embedding space. As shown in Figure 1(b), when an additional positive interaction lies far from previously observed positives, the constraints induced by the interaction set become difficult to satisfy simultaneously. To accommodate the new positive sample, the user embedding is forced to shift, which may weaken or even reverse its separation from other items. Consequently, conflicting proximity constraints are collapsed into a single representation, leading to unstable or locally inconsistent user–item relationships. This issue is further exacerbated in federated learning settings,

where each user embedding is inferred solely from local interactions and cannot benefit from how similar interaction patterns are resolved across other users, leaving such ambiguity fundamentally unaddressed.

This observation challenges the assumption that a single embedding can adequately represent a user. Under limited interaction evidence, forcing all observed behaviors into one representation imposes a restrictive interpretation of user intent. We therefore advocate a shift in perspective: **rather than treating user preference as a uniquely identifiable point, it should be modeled as a set of plausible representations consistent with the observed interactions**. As illustrated in Figure 1(c), this naturally leads to representing a user as a distribution in the embedding space rather than a single point. Such a representation spans multiple compatible regions, preserving alternative explanations of user behavior instead of committing to one arbitrarily. By retaining this ambiguity, the distributional view provides a more robust and faithful basis for user modeling.

Motivated by this perspective, we propose a novel federated recommendation framework that models each user representation as a distribution rather than a single embedding. The key idea is to adopt a generative modeling view, in which latent user preferences are treated as hidden variables that give rise to the observed user–item interactions. Under this view, the goal is not to identify one "best" user representation, but to characterize the set of latent representations that can reasonably explain the same interaction evidence.

Specifically, we instantiate this idea with a diffusion-based generative model, termed **FedDistRec** (**Fed**erated **Dist**ributional **Rec**ommendation), which constructs user representation distributions in the item embedding space. FedDistRec injects noise into representations derived from a user's interacted items and learns to denoise them into coherent user representations consistent with those items. By grounding both the corruption and recovery processes in item embeddings, the model captures user preferences through expressed item affinities rather than committing to a fixed latent vector. Moreover, within the diffusion generation process, FedDistRec incorporates both globally shared patterns and user-specific characteristics, enabling the learned user representations to encode common structure while remaining adaptive to individual users. This design provides a principled balance between generalization and personalization in federated recommendation. Our main contributions are summarized as follows:

- We identify a fundamental limitation of existing FRS in representing users with a single embedding, and advocate a principled shift toward distributional user representations, which offer a more expressive foundation under sparse and decentralized interactions.

- We propose a novel federated recommendation framework FedDistRec that learns user representation distributions via a diffusion-based generative model in the item embedding space, enabling the integration of globally shared patterns and client-specific characteristics for expressive and personalized user modeling.

- Extensive experiments demonstrate the superior performance of FedDistRec compared to competitive baselines. Additional comprehensive analyses confirm its compatibility, effectiveness, and practical applicability.

## 2. Related Work

### 2.1. Federated Recommender System

Driven by growing privacy demands, federated recommender systems (Li et al., 2026b; 2025; Zhang et al., 2025b; Sun et al., 2024; Wang et al., 2024b), which deploy recommendation models (Dang et al., 2025; 2026; Zhao et al., 2023; 2024a; Deng et al., 2026a) in a federated learning environment (Wang et al., 2024a; Chen et al., 2026; Kou et al., 2026), have emerged as a vital solution for delivering personalized recommendations while preserving user privacy. Early approaches such as FedMF (Chai et al., 2020) and FedNCF (Ammad-Ud-Din et al., 2019) adapt conventional recommendation models to federated environments (Yang et al., 2019), providing fundamental privacy safeguards and recommendation functionality. Methods like PFedRec (Zhang et al., 2023) and FedRAP (Li et al., 2024) further enhance performance through localized personalization strategies, refining user representation. However, these methods predominantly rely on deterministic embeddings that may inadvertently capture statistical biases (*e.g.,* prioritizing high-frequency items) rather than robust user intent (Deng et al., 2026b). By representing users with a single embedding, they neglect the diversity of user preferences and limit expressive power. To address this gap, we model each user's representation as a latent distributional structure, which captures the nuanced and multifaceted nature of individual user preferences.

### 2.2. Diffusion Models for Recommender Systems

Given their capacity to model complex distributions, diffusion models have recently been adopted as a powerful modeling component in recommender systems (Lin et al., 2024; Wei & Fang, 2025). Their application centers on three main areas: data augmentation and representation enhancement through auxiliary sample generation (Wang et al., 2023; Kotelnikov et al., 2023; Wu et al., 2023) and embedding denoising (Zhao et al., 2024b); diffusion as recommendation models that directly estimate user preferences (Yang et al., 2023; Ma et al., 2024; Wu et al., 2025; Luo et al., 2025); and content generation via synthesizing multimodal content

to match user taste (Rombach et al., 2022; Lugmayr et al., 2022). While diffusion models are inherently well suited for learning rich preference structures beyond point representations, existing approaches largely apply them to produce a single user embedding and are predominantly studied under centralized training settings. In this work, we leverage their generative capacity to explicitly learn user representations in a distributional form, and integrate this formulation into a federated learning framework, enabling both expressive preference modeling and privacy-preserving training over decentralized data.

## 3. Preliminary

This section provides the necessary background on federated recommendation and diffusion models that underpins the proposed methodology.

**General FRS Framework.** Let $\mathcal{U}$ and $\mathcal{I}$ denote the sets of users and items, with $|\mathcal{U}| = N$ and $|\mathcal{I}| = M$. A general FRS framework proceeds in iterative communication rounds between a central server and distributed clients. In each round, each client $u \in \mathcal{U}$ performs local updates on the shared recommendation model $\mathcal{F}_W$ using its private interaction set $\mathcal{R}_u$ (*e.g.,* clicked or purchased items). The server then aggregates these local updates to update the global model parameters, which are broadcast back to clients for the next round. This process corresponds to optimizing the following global objective:

$$\min_{W} \sum_{u=1}^{N} \alpha_u \mathcal{L}_u(W; \mathcal{R}_u) \tag{1}$$

where $\mathcal{L}_u$ denotes the local loss on client $u$, and $\alpha_u$ is the corresponding aggregation weight, typically determined by the relative size of the client's local interaction data.

**Diffusion Models.** Diffusion models (Ho et al., 2020; Nichol & Dhariwal, 2021) constitute a class of generative models widely used for modeling complex data distributions. They are formulated as a Markovian process consisting of a forward diffusion process and a reverse denoising process. Given an initial data sample $\mathbf{x}_0 \sim q(\mathbf{x}_0)$, with $\mathbf{x}_0 \in \mathbb{R}^d$, the forward process corrupts $\mathbf{x}_0$ over $T$ steps by injecting Gaussian noise according to a predefined variance schedule $\{\beta_t\}_{t=1}^{T}$, producing a sequence of noisy variables $\{\mathbf{x}_t\}_{t=1}^{T}$. Formally, the forward diffusion process is defined by the Gaussian transition:

$$q(\mathbf{x}_t|\mathbf{x}_{t-1}) = \mathcal{N}\left(\mathbf{x}_t; \sqrt{1-\beta_t}\,\mathbf{x}_{t-1}, \beta_t \mathbf{I}\right) \tag{2}$$

Using the reparameterization trick (Kingma & Welling, 2013), $\mathbf{x}_t$ can be written as

$$\mathbf{x}_t = \sqrt{\bar{\alpha}_t}\mathbf{x}_0 + \sqrt{1-\bar{\alpha}_t}\,\epsilon, \ \epsilon \sim \mathcal{N}(\mathbf{0}, \mathbf{I}) \tag{3}$$

where $\alpha_t = 1 - \beta_t$ and $\bar{\alpha}_t = \prod_{i=1}^{t} \alpha_i$.

The denoising process aims to reverse the forward diffusion by progressively removing noise and reconstructing the underlying data structure. When the clean sample $\mathbf{x}_0$ is available, the true posterior distribution $q(\mathbf{x}_{t-1} \mid \mathbf{x}_t, \mathbf{x}_0)$ admits a closed-form Gaussian expression:

$$q(\mathbf{x}_{t-1} \mid \mathbf{x}_t, \mathbf{x}_0) = \mathcal{N}\left(\mathbf{x}_{t-1}; \tilde{\mu}(\mathbf{x}_t, \mathbf{x}_0), \tilde{\beta}_t \mathbf{I}\right)$$
$$where \quad \tilde{\mu}(\mathbf{x}_t, \mathbf{x}_0) = \frac{\sqrt{\bar{\alpha}_{t-1}}\,\beta_t}{1-\bar{\alpha}_t}\mathbf{x}_0 + \frac{\sqrt{\alpha_t}(1-\bar{\alpha}_{t-1})}{1-\bar{\alpha}_t}\mathbf{x}_t,$$
$$\tilde{\beta}_t = \frac{1-\bar{\alpha}_{t-1}}{1-\bar{\alpha}_t}\beta_t$$
$$\tag{4}$$

This posterior provides an exact characterization of the reverse transition when $\mathbf{x}_0$ is known. However, at generation time, $\mathbf{x}_0$ is unavailable, rendering direct sampling from $q(\mathbf{x}_{t-1} \mid \mathbf{x}_t, \mathbf{x}_0)$ intractable. To address this, diffusion models introduce a parameterized reverse Markov chain $p_\theta$ to approximate the true posterior:

$$p_\theta(\mathbf{x}_{t-1}|\mathbf{x}_t) = \mathcal{N}\left(\mathbf{x}_{t-1}; \mu_\theta(\mathbf{x}_t, t), \Sigma_\theta(\mathbf{x}_t, t)\right) \tag{5}$$

Model training proceeds by minimizing the KL divergence between the learned reverse transition $p_\theta(\mathbf{x}_{t-1}|\mathbf{x}_t)$ and the true posterior $q(\mathbf{x}_{t-1} \mid \mathbf{x}_t, \mathbf{x}_0)$. A common and effective simplification fixes the covariance as $\Sigma_\theta(\mathbf{x}_t, t) = \tilde{\beta}_t \mathbf{I}$, while parameterizing the mean through a neural network $f_\theta(\mathbf{x}_t, t)$ that predicts the clean sample $\mathbf{x}_0$. Under this parameterization, the reverse mean is given by:

$$\mu_\theta(\mathbf{x}_t, t) = \frac{\sqrt{\bar{\alpha}_{t-1}}\beta_t}{1-\bar{\alpha}_t} f_\theta(\mathbf{x}_t, t) + \frac{\sqrt{\alpha_t}(1-\bar{\alpha}_{t-1})}{1-\bar{\alpha}_t}\mathbf{x}_t \tag{6}$$

With this formulation, the training objective reduces to a denoising mean squared error loss:

$$\min_{\theta} \ \mathbb{E}_{\mathbf{x}_0, t, \epsilon}\left[\left\|\mathbf{x}_0 - f_\theta(\mathbf{x}_t, t)\right\|_2^2\right] \tag{7}$$

where $t \sim \text{Uniform}(1, T)$, $\epsilon \sim \mathcal{N}(\mathbf{0}, \mathbf{I})$, and $\mathbf{x}_t$ is obtained via the forward diffusion process. After training, generation proceeds by iteratively sampling from $p_\theta(\mathbf{x}_{t-1} \mid \mathbf{x}_t)$, starting from Gaussian noise and ultimately producing samples from the learned data distribution.

## 4. Methodology

In this section, we introduce FedDistRec, a diffusion-based federated recommendation framework that learns user representation distributions in the item embedding space. The model constructs user representations through an iterative generative process conditioned on observed user–item interactions, capturing uncertainty and diversity in user preferences. This distributional formulation enables expressive and personalized user modeling by jointly encoding shared structure and individual characteristics. The overall framework is illustrated in Figure 2.

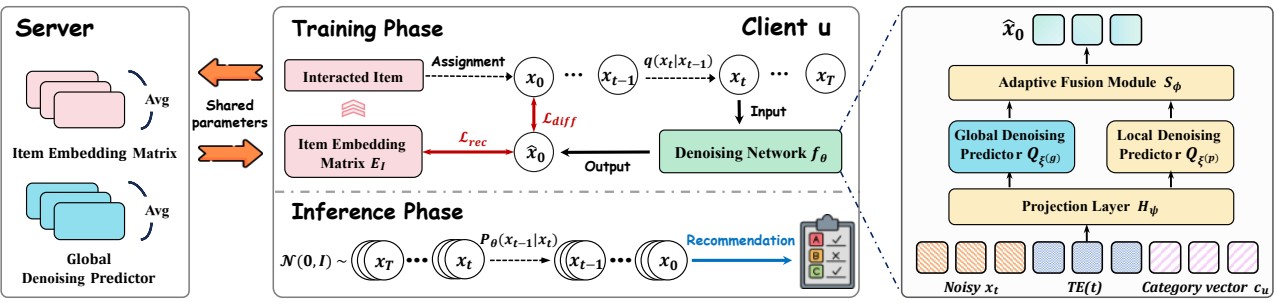

*Figure 2.* Overview of the proposed FedDistRec. Our framework adopts a diffusion-based approach to model user representations for federated recommendation, enabling clients to collaboratively learn shared structures while retaining personalization. The denoising function is explicitly decomposed into a globally shared component that captures population-level preference regularities and a locally maintained personalized component that models user-specific variations induced by heterogeneous interaction histories. During inference, FedDistRec iteratively denoises in the item embedding space to generate user representation distributions and ranks items based on their alignment with these distributions for final recommendation.

## 4.1. Diffusion-based User Modeling

We cast recommendation as a diffusion-based generative user modeling problem, in which user representations for downstream interaction prediction are constructed through a gradual denoising process rather than inferred as fixed latent vectors. In this formulation, $\mathbf{x}_0$ denotes a clean user representation constructed from an interacted item $i \in \mathcal{R}_u$ by retrieving its embedding from $E_{\mathcal{I}}$, while $\{\mathbf{x}_t\}_{t=1}^{T}$ are its noisy counterparts produced by a forward diffusion process. The reverse process is parameterized by $p_\theta(\mathbf{x}_{t-1}|\mathbf{x}_t)$, which progressively recovers coherent preference structure from noisy representations. By modeling users along this denoising trajectory, the diffusion framework captures variability in user preferences, yielding expressive representations that naturally support personalized recommendations.

**Semantic Conditioning for Reverse Diffusion.** In standard diffusion models, the reverse process $p_\theta(\mathbf{x}_{t-1}|\mathbf{x}_t)$ is conditioned only on the noisy representation $\mathbf{x}_t$ and the diffusion step $t$. However, in recommendation settings, user representations are inferred from sparse interaction histories, and at large diffusion steps, the item-derived preference information in $\mathbf{x}_t$ is largely overwhelmed by noise. In this regime, conditioning only on $(\mathbf{x}_t, t)$ leads to an ill-informed denoising process that cannot reliably recover preference-consistent user representations.

To address this limitation, we propose to explicitly augment the reverse diffusion process with semantic conditions derived from the user's interacted items. Specifically, we condition $p_\theta$ on item category semantic information that captures the high-level composition of a user's interactions, resulting in $f_\theta(\mathbf{x}_t, t, \mathbf{c}_u)$. By complementing the noisy representation $\mathbf{x}_t$, this semantic signal provides additional preference guidance and facilitates more stable denoising at large diffusion steps. For each user $u$, we construct a category-level semantic profile $\mathbf{c}_u \in \mathbb{R}^K$ based on the categories of

the items the user has interacted with:

$$\mathbf{c}_u = \left[ \pi_u^{(1)}, \ldots, \pi_u^{(k)}, \ldots \pi_u^{(K)} \right]^\top$$

$$where \quad \pi_u^{(k)} = \frac{\nu_u^{(k)}}{\sum_{j=1}^K \nu_u^{(j)}}, \quad \nu_u^{(k)} = \sum_{i \in \mathcal{R}_u} \sum_{c \in \mathcal{C}_i} \mathbb{I}(c = k)$$

$$(8)$$

where $\mathcal{R}_u$ denotes the set of items interacted with by user $u$; for each item $i \in \mathcal{R}_u$, $\mathcal{C}_i$ specifies its associated categories; $K$ is the total number of item categories; and $\mathbb{I}(\cdot)$ denotes the indicator function. The resulting vector $\mathbf{c}_u$ provides a compact semantic prior over item categories, which can help guide the diffusion process toward preference-consistent user representations.

**Conditional Input Encoding.** We parameterize the reverse diffusion process by a denoising function conditioned on the noisy user representation $\mathbf{x}_t$, the diffusion step $t$, and the user-specific semantic condition $\mathbf{c}_u$. Both $\mathbf{x}_t \in \mathbb{R}^d$ and $\mathbf{c}_u \in \mathbb{R}^K$ are vector-valued inputs, while the diffusion step $t$ is encoded using a fixed sinusoidal time embedding denoted as $TE(t)$, following standard diffusion practice (Rombach et al., 2022; Ho & Salimans, 2022). We therefore project all inputs into a shared latent space:

$$\mathbf{h}_t = H_\psi(\mathbf{x}_t, TE(t), \mathbf{c}_u) \qquad (9)$$

where $H_\psi$ denotes a projection layer (*e.g.,* concatenation, MLP, or attention). This encoding integrates noisy preference signals, diffusion-stage information, and semantic context into a unified representation that serves as the input to the denoising network.

**Global–Personalized Denoising Parameterization.** Given the conditioned latent representation $\mathbf{h}_t$, we decompose the denoising function into a global component shared across all clients and a personalized component specific to each user. The global component captures population-level preference patterns that are consistent across decentralized clients,

while the personalized component accounts for user-specific deviations induced by heterogeneous interaction histories. Formally, we define two parallel denoising predictors:

$$\hat{\mathbf{x}}_0^{(g)} = Q_{\xi^{(g)}}(\mathbf{h}_t), \quad \hat{\mathbf{x}}_0^{(p)} = Q_{\xi^{(p)}}(\mathbf{h}_t) \qquad (10)$$

where $Q_{\xi^{(g)}}$ is parameterized by globally shared parameters and updated via federated aggregation, and $Q_{\xi^{(p)}}$ is parameterized by user-specific parameters and trained locally without sharing. This decomposition enables the model to leverage federated knowledge for robust denoising while retaining flexibility to model personalized preference signals.

**Adaptive Fusion for Denoising Prediction.** To combine the global and personalized denoising outputs, we introduce an adaptive fusion module that produces the final prediction by integrating both views. The fusion module learns how to weight and merge the two predictions, allowing the model to leverage shared patterns while retaining user-specific adjustments:

$$\hat{\mathbf{x}}_0 = S_\phi(\hat{\mathbf{x}}_0^{(g)}, \hat{\mathbf{x}}_0^{(p)}) \qquad (11)$$

In practice, $S_\phi$ can be implemented through transformation matrices, gating mechanisms, or attention-based weighting to generate the fused embedding.

## 4.2. Joint Training for User Modeling and Recommendation

In each client, the recommender system is composed of two coupled components: (1) a diffusion-based user representation learner, and (2) a downstream prediction head that converts the learned representation into recommendation scores. Training starts by constructing a clean user embedding $\mathbf{x}_0$ from the user's interaction history using the shared item embedding matrix $E_{\mathcal{I}}$. A diffusion step $t$ is then sampled uniformly, and the corresponding noisy representation $\mathbf{x}_t$ is generated via the forward diffusion process. The reverse denoising network $f_\theta(\mathbf{x}_t, t, \mathbf{c}_u)$ produces a reconstruction $\hat{\mathbf{x}}_0$, and the diffusion learner is optimized by minimizing the reconstruction error:

$$\mathcal{L}_{diff} = \mathbb{E}_{\mathbf{x}_0, t, \epsilon}\left[\|\mathbf{x}_0 - f_\theta(\mathbf{x}_t, t, \mathbf{c}_u)\|_2^2\right] \qquad (12)$$

The reconstructed embedding $\hat{\mathbf{x}}_0$ is used as the user representation in the recommendation task. Specifically, the prediction head $g_\omega$ computes preference scores between $\hat{\mathbf{x}}_0$ and item embeddings, and a standard ranking loss (*e.g.,* BPR loss or cross-entropy loss) is minimized to align predicted rankings with observed interactions.

To support federated learning, we partition model parameters into shared and private sets. The item embedding matrix $E_{\mathcal{I}}$ and the global denoising predictor $Q_{\xi^{(g)}}$ are shared across clients, since they capture common item semantics and general denoising patterns. All remaining parameters

are kept private to preserve client-specific preference modeling. This design enables global knowledge transfer while maintaining personalization.

The overall local objective for client $u$ combines the diffusion reconstruction loss and the recommendation loss:

$$\mathcal{L}_u = \mathcal{L}_{rec}(E_{\mathcal{I}}, \omega; \mathcal{R}_u) + \lambda\,\mathcal{L}_{diff}(\theta; E_{\mathcal{I}}) \qquad (13)$$

where $\theta = \{\psi, \xi^{(g)}, \xi^{(p)}, \phi\}$ denotes the diffusion model parameters (including the encoding network $\psi$, the global and personalized denoising branches $\xi^{(g)}$ and $\xi^{(p)}$, and the adaptive fusion module $\phi$). The scalar $\lambda$ controls the trade-off between recommendation accuracy and diffusion reconstruction. During federated optimization, the shared parameters are aggregated across clients, while the remaining parameters are kept local to preserve user-specific modeling capacity. See Appendix A.1 for details.

## 4.3. Local Inference and Recommendation

**User Embedding Generation with Multiple Samples.** To capture the uncertainty of user preferences and reduce the variance of a single reconstructed embedding, we generate multiple candidate user representations through parallel reverse diffusion trajectories. For user $u$, the client initializes $N_s$ independent Gaussian noise vectors $\{\mathbf{x}_T^{(s)} \sim \mathcal{N}(\mathbf{0}, \mathbf{I})\}_{s=1}^{N_s}$. At each step $t$, the denoising network predicts a clean embedding:

$$\hat{\mathbf{x}}_0^{(s)}(t) = f_\theta(\mathbf{x}_t^{(s)}, t, \mathbf{c}_u) \qquad (14)$$

where $\hat{\mathbf{x}}_0^{(s)}(t)$ is the network's estimate of the clean user embedding given the current noisy state. Using this estimate, we compute the reverse transition mean:

$$\mu_t^{(s)} = \frac{\sqrt{\bar{\alpha}_{t-1}}\beta_t}{1 - \bar{\alpha}_t}\hat{\mathbf{x}}_0^{(s)}(t) + \frac{\sqrt{\alpha_t}(1 - \bar{\alpha}_{t-1})}{1 - \bar{\alpha}_t}\mathbf{x}_t^{(s)} \qquad (15)$$

Then the next noisy state is sampled as:

$$\mathbf{x}_{t-1}^{(s)} \sim \mathcal{N}\left(\mu_t^{(s)}, \frac{1 - \bar{\alpha}_{t-1}}{1 - \bar{\alpha}_t}\beta_t\mathbf{I}\right) \qquad (16)$$

After completing $T$ steps, the final $\mathbf{x}_0^{(s)}$ is obtained, which can be treated as the user embeddings for recommendation.

**Recommendation Scoring with Ensemble User Embeddings.** Once $N_s$ denoised user embeddings $\{\mathbf{x}_0^{(s)}\}_{s=1}^{N_s}$ are obtained, we compute preference scores for each candidate item $i \in \mathcal{I}$ using the recommendation prediction head $g_\omega$ (*e.g.,* dot product, cosine similarity, or a neural network):

$$s_{ui}^{(s)} = g_\omega\left(\mathbf{x}_0^{(s)}, E_{\mathcal{I}}(i)\right) \qquad (17)$$

The final score for item $i$ is then obtained by averaging these $N_s$ scores:

$$\bar{s}_{ui} = \frac{1}{N_s}\sum_{s=1}^{N_s} s_{ui}^{(s)} \qquad (18)$$

*Table 1.* Overall performance comparison of FedDistRec and baselines. H@10, N@10, R@10, and P@10 denote HR@10, NDCG@10, Recall@10, and Precision@10, respectively. Bold indicates the best result, underline marks the second best result.

| Method | | Toys | | | | Health | | | | Apps | | | | KuaiSAR | | | |
|---|---|---|---|---|---|---|---|---|---|---|---|---|---|---|---|---|---|---|
| | | H@10 | N@10 | R@10 | P@10 | H@10 | N@10 | R@10 | P@10 | H@10 | N@10 | R@10 | P@10 | H@10 | N@10 | R@10 | P@10 |
| *Box-embedding Methods* | | | | | | | | | | | | | | | | | |
| HCUR | | 0.2949 | 0.0660 | 0.0957 | 0.0338 | 0.3277 | 0.0752 | 0.0964 | 0.0406 | 0.3683 | 0.0768 | 0.1041 | 0.0453 | 0.8734 | 0.2384 | 0.1076 | 0.2269 |
| LCD-UC | | 0.3154 | 0.0761 | 0.1040 | 0.0378 | 0.3406 | 0.0689 | 0.0952 | 0.0438 | 0.3583 | 0.0740 | 0.1015 | 0.0438 | 0.8225 | 0.1809 | 0.0835 | 0.1788 |
| *Diffusion-based Methods* | | | | | | | | | | | | | | | | | |
| DDRM | | 0.3063 | 0.0660 | 0.0976 | 0.0361 | 0.3375 | 0.0699 | 0.0961 | 0.0426 | 0.3549 | 0.0718 | 0.0987 | 0.0434 | 0.8214 | 0.1779 | 0.0827 | 0.1773 |
| GCDR | | 0.2978 | 0.0662 | 0.0967 | 0.0354 | 0.3545 | 0.0748 | 0.1010 | 0.0453 | 0.4313 | 0.0961 | 0.1296 | 0.0551 | 0.8775 | 0.2326 | 0.1057 | 0.2219 |
| *Federated Recommendation Methods* | | | | | | | | | | | | | | | | | |
| FedMF | vanilla | 0.2983 | 0.0646 | 0.0978 | 0.0353 | 0.3426 | 0.0721 | 0.0976 | 0.0444 | 0.3450 | 0.0703 | 0.0952 | 0.0419 | 0.8188 | 0.1786 | 0.0830 | 0.1788 |
| | w/ ours | 0.3139 | 0.0665 | 0.0992 | 0.0366 | 0.3608 | 0.0770 | 0.1053 | 0.0473 | 0.3560 | 0.0724 | 0.0996 | 0.0434 | 0.8287 | 0.1835 | 0.0856 | 0.1828 |
| FedNCF | vanilla | 0.3125 | 0.0670 | 0.0982 | 0.0369 | 0.3559 | 0.0866 | 0.1110 | 0.0443 | 0.3697 | 0.0779 | 0.1049 | 0.0456 | 0.8748 | 0.2353 | 0.1071 | 0.2246 |
| | w/ ours | 0.3432 | **0.0795** | 0.1151 | 0.0412 | 0.4512 | **0.1422** | 0.1412 | 0.0588 | 0.3933 | 0.0830 | 0.1147 | 0.0493 | 0.9417 | 0.3661 | 0.1733 | 0.3484 |
| PFedRec | vanilla | 0.3066 | 0.0645 | 0.0979 | 0.0351 | 0.3395 | 0.0713 | 0.0954 | 0.0435 | 0.3481 | 0.0711 | 0.0973 | 0.0427 | 0.8231 | 0.1786 | 0.0837 | 0.1793 |
| | w/ ours | **0.3459** | 0.0756 | **0.1152** | **0.0417** | **0.4634** | 0.1331 | **0.1478** | **0.0610** | **0.6392** | **0.2202** | **0.2594** | **0.1052** | **0.9426** | **0.3662** | **0.1753** | **0.3510** |
| FedRAP | vanilla | 0.3034 | 0.0643 | 0.0968 | 0.0349 | 0.3423 | 0.0714 | 0.0988 | 0.0438 | 0.3422 | 0.0704 | 0.0966 | 0.0423 | 0.8119 | 0.1795 | 0.0825 | 0.1788 |
| | w/ ours | 0.3194 | 0.0718 | 0.1039 | 0.0386 | 0.3536 | 0.0747 | 0.1023 | 0.0449 | 0.3548 | 0.0727 | 0.0986 | 0.0430 | 0.8268 | 0.1823 | 0.0851 | 0.1824 |

Items are ranked according to $\bar{s}_{ui}$ for recommendation. See the complete algorithm in Appendix A.2.

### 4.4. Discussion on Practical Adaptability

**Deployment guidelines.** FedDistRec is designed for modularity and compatibility. The denoising network adopts a resource-aware architecture, in which each component admits alternative implementations, enabling scalability across diverse deployment settings. As a plug-in module, FedDistRec can be seamlessly integrated into existing federated recommendation pipelines without modifying the base model architecture; the denoising network is incorporated as an auxiliary component, and training requires only the introduction of an additional loss term.

**Security analysis.** By retaining personalized parameters locally, FedDistRec minimizes user-specific information in shared updates, reducing the risk of gradient inversion and attribute inference attacks. Compared to standard federated recommendation models, it offers comparable privacy protection. Additionally, FedDistRec can integrate privacy-enhancing techniques such as Differential Privacy (Choi et al., 2018) and Homomorphic Encryption (Acar et al., 2018) to further enhance security. See the experimental section for privacy evaluation.

### 4.5. Convergence Analysis

We analyze the convergence of the global parameters in FedDistRec, which optimizes a diffusion-based distributional user modeling objective under federated learning. The parameter space $W = \{E_\mathcal{I}, \omega, \theta\}$ is decomposed into globally shared parameters $W_g$ and client-specific personalized parameters $W_p$. Since only $W_g$ participates in server aggregation, our analysis focuses on its convergence. Under standard assumptions in nonconvex federated optimization, we derive the following convergence bound:

$$\min_{\tau \in \{0,\ldots,R-1\}} \mathbb{E}\|\nabla_{W_g} F(W^{(\tau)})\|^2 \le \underbrace{\frac{2(F(W^{(0)}) - F^\star)}{\eta_l H R}}_{\text{Initial Gap}}$$
$$+ \underbrace{\mathcal{O}(\eta_l H(\sigma_g^2 + \zeta^2)/S)}_{\text{Noise \& Heterogeneity}} + \underbrace{\mathcal{O}(L^2 \eta_l H(G_g^2 + \sigma_g^2))}_{\text{Client Drift}}$$

(19)

This bound indicates that, even with the additional stochasticity induced by diffusion-based user representation generation, the optimization of globally shared parameters remains stable and converges to a stationary point up to controllable errors. Specifically, the initial gap diminishes with the number of communication rounds $R$, while the effects of stochastic noise and client heterogeneity can be alleviated by sampling more clients per round. The client drift term, arising from multiple local updates, further highlights the need to balance the number of local steps $H$ and the local learning rate $\eta_l$. Detailed proofs are provided in Appendix B.

## 5. Experiment

### 5.1. Experimental Details

We evaluate FedDistRec on four real-world benchmark datasets: Toys, Health, and Apps from Amazon (Hou et al., 2024), and KuaiSAR from Kuaishou (Sun et al., 2023). These datasets span diverse domains, sparsity levels, and scales, consisting of user–item interactions accompanied by item-side categorical metadata. We partition each dataset into training, validation, and test sets with a 7:1:2 ratio, excluding users with fewer than 10 interactions in the Amazon datasets or fewer than 50 interactions in KuaiSAR. Table 2 summarizes the dataset statistics. Following prior work (He et al., 2017; Zhang et al., 2023), we sample four negative instances per positive instance. For evaluation, we adopt four ranking-based metrics for top-$K$ recommenda-

*Table 2.* Dataset statistics.

| Dataset | #Users | #Items | #Interactions | Sparsity | #Category |
|---|---|---|---|---|---|
| Toys | 912 | 857 | 18,304 | 97.66% | 17 |
| Health | 2,160 | 1,233 | 54,046 | 97.97% | 9 |
| Apps | 5,080 | 3,221 | 122,528 | 99.25% | 29 |
| KuaiSAR | 4,400 | 6,882 | 517,381 | 98.29% | 37 |

tion: Hit Rate (HR@$K$), Normalized Discounted Cumulative Gain (NDCG@$K$), Recall@$K$ and Precision@$K$. All experiments are implemented in PyTorch and run on three NVIDIA GeForce RTX 3090 GPUs, with results averaged over five independent runs. Additional experimental details are provided in Appendix C.

## 5.2. Baselines

To thoroughly evaluate our approach, we compare it against three categories of baselines:

*(1) Box-embedding Methods.* We include them as baselines since they represent user preferences as regions in the embedding space, a design that is conceptually related to our goal of capturing diverse user preferences:

- **HCUR** (Zhang et al., 2021) represents user as hypercuboids and scores items via compositional distances.
- **LCD-UC** (Wu et al., 2024) encodes both users and items as hypercuboids and computes similarity through an attention mechanism.

*(2) Diffusion-based Methods.* We additionally compare against two diffusion-based recommenders that employ generative diffusion models for preference modeling, excluding approaches that use diffusion solely for data augmentation, as these methods are most closely aligned with the generative mechanism underlying our framework:

- **DDRM** (Zhao et al., 2024b) enhances embedding robustness via a denoising diffusion process.
- **GCDR** (Wu et al., 2025) captures diverse user interests with a conditional diffusion framework that models each user via multiple distributions.

To ensure a fair comparison, we adapt the above centralized methods to the federated setting by retaining user-specific parameters locally while aggregating the remaining parameters across clients.

*(3) Federated Recommendation Methods.* We also selecte four state-of-the-art federated recommendation methods for a more comprehensive comparison. By integrating our framework into these backbones, we are able to directly assess its plug-and-play compatibility and measure the resulting performance enhancement:

- **FedMF** (Chai et al., 2020) employs matrix factorization in a federated setting.

- **FedNCF** (Ammad-Ud-Din et al., 2019) adopts a neural collaborative filtering framework to learn shared item embeddings and a global prediction function.

- **PFedRec** (Zhang et al., 2023) enables fine-grained user-level personalization through a dual personalization mechanism.

- **FedRAP** (Li et al., 2024) maintains both global and personalized item embeddings per client, balancing shared and individual characteristics through a tailored loss function.

## 5.3. Overall Comparison

Table 1 summarizes the performance of all methods on four datasets in terms of HR@10, NDCG@10, Recall@10, and Precision@10. The key findings are as follows:

*(1) Our distributional user modeling offers greater representational flexibility than region-based approaches.* Box-embedding methods encode each user as a single fixed region in the embedding space, yielding a deterministic preference representation. In contrast, our diffusion-based approach models user representations as a distribution conditioned on interaction history, allowing multiple plausible embeddings to be generated for the same user. This additional flexibility enables the model to better accommodate variability in user preferences and leads to more effective ranking performance in practice. *(2) Diffusion models benefit from explicit global–personalized decomposition in a federated setting.* Compared with existing diffusion-based recommenders, which either denoise user representations into a single embedding (DDRM) or maintain multiple latent distributions within each client (GCDR), our approach achieves stronger recommendation performance by explicitly separating global and personalized diffusion predictors. This design retains user-specific adaptation while controlling client-side complexity, leading to more reliable preference modeling in federated settings. *(3) Our method shows consistent performance gains across different backbone models.* Across different FRS backbones, incorporating our FedDistRec into the user modeling stage consistently improves recommendation accuracy. The core architectures of these backbones remain unchanged during integration, suggesting that the observed gains stem from enhanced user representations rather than from backbone-specific modeling assumptions. The consistent improvements across architectures further demonstrate the general applicability of the proposed approach.

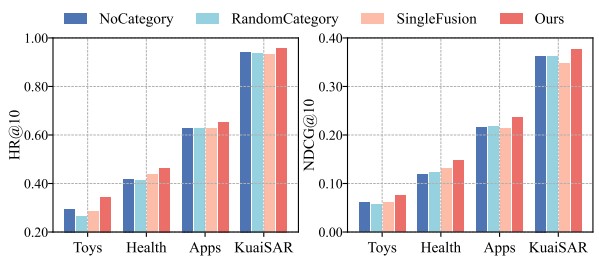

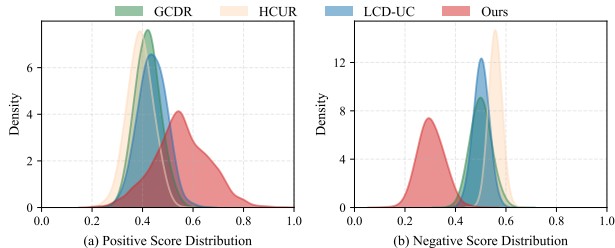

*Figure 3.* Ablation Study for Category Semantic Conditioning and Global–Personalized Denoising Parameterization.

*Figure 4.* Distributions of prediction scores on the Toys Dataset.

## 5.4. Ablation Study

This section investigates the contribution of key design choices in the proposed framework through targeted ablation experiments. In particular, we examine (i) the effectiveness of incorporating **category semantic conditioning** into the diffusion process, and (ii) the role of the **global–personalized denoising parameterization** in balancing shared behavioral patterns and user-specific preferences. By systematically removing or altering these components while keeping other settings unchanged, the ablation results aim to clarify how each element influences representation quality and downstream recommendation performance. We evaluate our method by incorporating it into PFedRec and report the HR@10, NDCG@10 across all datasets. Complete results can be found in Appendix D.1.

**Effect of Category Semantic Conditioning.** We evaluate the role of category semantic conditioning by comparing the full model with two ablated variants: one that removes category semantics from the conditional input (**NoCategory**), and another that replaces them with randomly initialized vectors (**RandomCategory**). As shown in Figure 3, our model consistently outperforms both variants across all metrics. This indicates that the improvement depends on category semantics that accurately reflect users' interaction distributions over item categories, rather than on the presence of category-related signals alone.

**Global–Personalized Denoising Parameterization.** We examine the effect of separating global and personalized denoising parameters by comparing the full model with a variant that removes the personalized denoising branch and relies solely on shared denoising parameters (**SingleFusion**). As shown in Figure 3, our model consistently outperforms SingleFusion across all evaluation metrics. These results demonstrate that modules dedicated to individualized modeling play a critical role in capturing distinctive user characteristics. By jointly modeling globally shared behavioral patterns and user-specific refinements, the proposed dual-view design yields more accurate and adaptive recommendations.

## 5.5. In-depth Analysis of Distribution-Induced Ranking Behavior

To further examine the effectiveness and necessity of modeling user representations as distributions, this section provides an in-depth analysis of how such representations influence model predictions. Rather than focusing solely on overall recommendation accuracy, we analyze the resulting prediction behaviors to understand whether learning multiple user representations leads to more reliable discrimination between positive and negative items. We conduct a comparative analysis against representative baselines that also aim to capture multiple aspects of user preference, including box-embedding methods (HCUR, LCD-UC) and a diffusion-based recommender (GCDR).

**Predicted Score Distributions for Positive and Negative Items.** We visualize the empirical distributions of prediction scores for positive and negative items. For each user, we compute the mean score of positive and negative test items separately, and then aggregate them across users to obtain two score distributions. An effective recommender should assign higher scores to positive items than to negative ones, resulting in two well-separated distributions.

As shown in Figure 4, our method exhibits a score distribution that is closer to the ideal separation: the scores of positive items peak around 0.6 with most probability mass above 0.5, whereas negative items are largely concentrated below 0.4. In contrast, all baselines produce substantially overlapping distributions for positive and negative items, both centered in the range of 0.4 to 0.5. These results suggest that our approach produces more discriminative and better-calibrated prediction scores, which aligns with the observed improvements in recommendation performance.

**User-Grouped Ranking Violation Analysis.** Beyond global score distributions, we further investigate whether the learned user representation distributions can reliably preserve the relative ordering between positive and negative items at the user level. We focus on the frequency of ranking violations, defined as instances where a user's positive items receive lower predicted scores than negative ones. A lower violation rate indicates more consistent assignment of

*Table 3.* User-level ranking violation rates across interaction regimes on the Toys dataset. Users are grouped by interaction frequency and interaction diversity using quartile-based splits, where Q1 denotes the lowest quartile and Q4 denotes the highest. The best result is shown in **bold**. The green values indicate the relative reduction compared to the second-best result for this metric (lower is better).

| Ranking Violation Rate ($\downarrow$) | Interaction Frequency (Quartiles) | | | | Interaction Diversity (Quartiles) | | | |
|---|---|---|---|---|---|---|---|---|
| | Q1 (Lowest) | Q2 | Q3 | Q4 (Highest) | Q1 (Lowest) | Q2 | Q3 | Q4 (Highest) |
| HCUR | 0.5190 | 0.5107 | 0.5007 | 0.4724 | 0.4922 | 0.5049 | 0.5108 | 0.4970 |
| LCD-UC | 0.5277 | 0.5129 | 0.5264 | 0.5149 | 0.5116 | 0.5218 | 0.5290 | 0.5206 |
| GCDR | 0.5361 | 0.5174 | 0.5375 | 0.5098 | 0.5225 | 0.5252 | 0.5265 | 0.5264 |
| **Ours** | **0.4786** (7.78%) | **0.4861** (4.82%) | **0.4884** (2.46%) | **0.4595** (2.73%) | **0.4630** (5.93%) | **0.4822** (4.50%) | **0.4879** (4.48%) | **0.4769** (4.04%) |

*Table 4.* Performance of integrating LDP into our proposed Fed-DistRec under different noise scales $\delta$.

| Dataset | Metric | $\delta=0$ | $\delta=0.1$ | $\delta=0.2$ | $\delta=0.3$ | $\delta=0.4$ | $\delta=0.5$ |
|---|---|---|---|---|---|---|---|
| | H@10 | 0.3459 | 0.3425 | 0.3366 | 0.3202 | 0.3070 | 0.2958 |
| Toys | N@10 | 0.0756 | 0.0748 | 0.0716 | 0.0703 | 0.0659 | 0.0620 |
| | R@10 | 0.1152 | 0.1119 | 0.1090 | 0.1082 | 0.0997 | 0.0995 |
| | P@10 | 0.0417 | 0.0404 | 0.0402 | 0.0365 | 0.0352 | 0.0309 |
| | H@10 | 0.4634 | 0.4314 | 0.4114 | 0.3866 | 0.3606 | 0.3514 |
| Health | N@10 | 0.1331 | 0.1175 | 0.1057 | 0.0897 | 0.0808 | 0.0789 |
| | R@10 | 0.1478 | 0.1236 | 0.1069 | 0.0986 | 0.0918 | 0.0814 |
| | P@10 | 0.0610 | 0.0583 | 0.0554 | 0.0497 | 0.0454 | 0.0378 |
| | H@10 | 0.6392 | 0.6116 | 0.5943 | 0.5828 | 0.5761 | 0.5731 |
| Apps | N@10 | 0.2202 | 0.2108 | 0.2067 | 0.1907 | 0.1872 | 0.1792 |
| | R@10 | 0.2594 | 0.2484 | 0.2291 | 0.2197 | 0.2146 | 0.2017 |
| | P@10 | 0.1052 | 0.1036 | 0.1004 | 0.0945 | 0.0908 | 0.0893 |
| | H@10 | 0.9426 | 0.8525 | 0.8218 | 0.8168 | 0.8036 | 0.7918 |
| KuaiSAR | N@10 | 0.3662 | 0.2048 | 0.1767 | 0.1752 | 0.1743 | 0.1607 |
| | R@10 | 0.1753 | 0.0947 | 0.0828 | 0.0821 | 0.0811 | 0.0768 |
| | P@10 | 0.3510 | 0.2012 | 0.1769 | 0.1743 | 0.1702 | 0.1653 |

higher relevance scores to truly preferred items, reflecting a more faithful encoding of user–item relationships.

To account for heterogeneity in user behavior, users are partitioned into four subgroups according to the upper and lower quartiles of interaction frequency and interaction diversity. Formally, interaction frequency is defined as the number of unique items interacted with by a user, while interaction diversity measures the number of distinct item categories covered by the user's interactions in the training data. For each subgroup, we compute the average user-level ranking violation rate. As shown in Table 3, our model consistently achieves the lowest violation rates across all user groups. The advantage is particularly pronounced in challenging regimes, such as low-frequency and low-diversity users, where limited interaction signals make accurate user representation learning especially difficult. Even under these adverse conditions, our approach maintains clear score separation between positive and negative items, indicating that the learned user representation distributions remain stable and informative.

### 5.6. Hyper-Parameter Analysis

This subsection examines the influence of key hyperparameters. **(i) Regularization coefficient** $\lambda$. The coefficient $\lambda$ balances the reconstruction objective and the learning of user preference signals. Empirically, a small value weakens reconstruction, leading to under-trained latent representations,

while an excessively large value overemphasizes reconstruction and suppresses preference discrimination. Across all datasets, setting $\lambda = 0.1$ provides a stable balance and consistently yields strong performance. **(ii) Number of inference samples** $N_s$. The number of inference samples determines how accurately the user representation distribution is approximated at inference time. Increasing $N_s$ improves performance by reducing approximation noise. In our experiments, $N_s = 5$ is sufficient to achieve reliable performance across datasets, offering an effective balance between accuracy and computational efficiency. Full results can be found in Appendix D.2.

### 5.7. Privacy Protection-Enhanced FedDistRec

To enhance the privacy guarantees of the proposed Fed-DistRec, we incorporate local differential privacy (LDP) by injecting Laplacian noise into all shared parameters before they are transmitted to the server. Specifically, we conduct experiments across five noise intensity levels by adjusting the noise scale $\delta \in \{0.1, 0.2, 0.3, 0.4, 0.5\}$ with unit sensitivity, corresponding to privacy budgets $\epsilon \in \{10, 5, 3.3, 2.5, 2\}$. As summarized in Table 4, stronger noise improves privacy protection but also results in a progressive degradation of model accuracy. In practice, an intermediate noise setting can strike a viable balance, offering robust privacy while maintaining acceptable recommendation performance.

## 6. Conclusion

In federated recommender systems, learning a single user embedding from sparse local interactions is prone to ambiguity and unstable user–item relations. We challenge this deterministic assumption and propose FedDistRec, a diffusion-based framework that models each user as a distribution over the item embedding space. By generating user representations through a noise-and-denoise process, the method preserves multiple plausible preference explanations while integrating shared global patterns and user-specific traits. Extensive experiments demonstrate that FedDistRec consistently improves recommendation accuracy and robustness, confirming the practical value of distributional user modeling in decentralized recommendation.

## Acknowledgements

Chunxu Zhang, Weipeng Zhang, Zhiheng Xue and Bo Yang are supported by the National Natural Science Foundation of China under Grant Nos. U22A2098, 62206105 and 62202200; the Major Science and Technology Development Plan of Jilin Province under Grant No.20240212003GX, the Major Science and Technology Development Plan of Changchun under Grant No.2024WX05.

## Impact Statement

This work advances privacy-preserving personalized recommendations by leveraging federated learning, offering a promising direction for enhancing user data protection and supporting decentralized AI applications.

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

## A. Algorithm

### A.1. Optimization Algorithm for the Training Phase

The optimization is performed via an iterative server-client procedure, as outlined in Algorithm 1. **Prior to the federated optimization,** the server first conducts a unified initialization of the globally shared parameters, which include the item embedding matrix $E_{\mathcal{I}}^{(1)}$ and the global denoising predictor $Q_{\xi^{(g)}}^{(1)}$. **In each communication round $r$,** the server randomly selects a subset of clients $B^{(r)}$ with sampling ratio $\beta$, and broadcasts the latest shared parameters $E_{\mathcal{I}}^{(r)}$ and $Q_{\xi^{(g)}}^{(r)}$ to them. Each selected client $u \in B^{(r)}$ initializes its local models using the shared parameters from the server and its personal parameters retained from the previous round. The client then updates these parameters based on local interaction data $\mathcal{R}_u$ by optimizing the objective defined in Eq.(13), and subsequently uploads only the updated shared components to the server. Upon receiving the updated parameters $\{E_{\mathcal{I}}^u\}_{u \in B^{(r)}}$ and $\{Q_{\xi^{(g)}}^u\}_{u \in B^{(r)}}$, the server aggregates them to produce the new shared parameters $E_{\mathcal{I}}^{(r+1)}$ and $Q_{\xi^{(g)}}^{(r+1)}$ for the next communication round.

### A.2. Algorithm for the Inference Phase

The inference phase performs a local, client-side diffusion process to generate personalized recommendations for user $u$ without requiring communication with the server. As presented in Algorithm 2, the procedure begins by sampling a set of $N_s$ latent noise vectors $\mathbf{x}_T$ from a standard Gaussian distribution $\mathcal{N}(\mathbf{0}, \mathbf{I})$, alongside the extraction of the user-specific semantic condition $\mathbf{c}_u$ using Eq. (8). Subsequently, a reverse diffusion procedure is executed iteratively from timestep $t = T$ down to $t = 1$, with each step refining the latent representations via the update rule defined in Eq. (16). The resulting denoised vectors are interpreted as user-aligned embedding candidates. Recommendation scores are then computed by applying Eq. (17) and Eq. (18).

## B. Convergence Proofs

In this section, we analyze FedDistRec's convergence in the standard nonconvex federated setting.

**Problem Formulation.** Let $W_g$ denote the shared parameters (*e.g.,* item embedding matrix, global denoising predictor) and $W_p$ denote the client-resident personalized parameters. The global objective is:

$$F(W) = \sum_{u \in \mathcal{U}} \alpha_u F_u(W), \quad \text{with} \quad W := \big(E_{\mathcal{I}}, \omega, \theta\big) = \big(W_g, W_p\big)$$

where $\alpha_u \geq 0$ is the weight of client $u$, and the local objective $F_u$ is as follows:

$$F_u(W) = \mathcal{L}_{rec} + \lambda \mathcal{L}_{diff}$$

We provide a convergence analysis under the following standard assumptions:

(A1) $L$-**Smoothness.** Each $F_u$ is $L$-smooth with respect to the shared parameters $W_g$:

$$\|\nabla_{W_g} F_u(W_g^{(1)}, W_p) - \nabla_{W_g} F_u(W_g^{(2)}, W_p)\| \leq L \|W_g^{(1)} - W_g^{(2)}\|$$

(A2) **Bounded Gradient.** $\|\nabla_{W_g} F_u(W)\| \leq G_g$ for all $u$.

(A3) **Bounded Stochastic Variance.** Stochastic gradients are unbiased and satisfy

$$\mathbb{E}[g_{g,u}(W)] = \nabla_{W_g} F_u(W), \quad \mathbb{E}\|g_{g,u}(W) - \nabla_{W_g} F_u(W)\|^2 \leq \sigma_g^2$$

(A4) **Bounded Global Heterogeneity.** Local gradients satisfy

$$\mathbb{E}_u \|\nabla_{W_g} F_u(W) - \nabla_{W_g} F(W)\|^2 \leq \zeta^2$$

**Proof.** Let client $u$ perform $H$ local steps with learning rate $\eta_l$:

$$W_{g,u}^{(\tau,h+1)} = W_{g,u}^{(\tau,h)} - \eta_l g_{g,u}(W_u^{(\tau,h)})$$

so that

$$W_{g,u}^{(\tau,H)} - W_g^{(\tau)} = -\eta_l \sum_{h=0}^{H-1} g_{g,u}(W_u^{(\tau,h)})$$

Taking expectation and applying $\|\sum_i a_i\|^2 \leq n \sum_i \|a_i\|^2$, together with (A2)-(A3), gives

$$\mathbb{E}\|W_{g,u}^{(\tau,H)} - W_g^{(\tau)}\|^2 \leq H^2 \eta_l^2 (G_g^2 + \sigma_g^2)$$

Define the global update at round $\tau$ as

$$\Delta_\tau = W_g^{(\tau+1)} - W_g^{(\tau)} = \frac{1}{|S_\tau|} \sum_{u \in S_\tau} (W_{g,u}^{(\tau,H)} - W_g^{(\tau)})$$

By $L$-smoothness (A1),

$$F(W^{(\tau+1)}) \leq F(W^{(\tau)}) + \langle \nabla_{W_g} F(W^{(\tau)}), \Delta_\tau \rangle + \frac{L}{2}\|\Delta_\tau\|^2$$

Decompose the inner product using the client drift

$$\text{drift}_u := W_{g,u}^{(\tau,H)} - W_g^{(\tau)} + \eta_l H \nabla_{W_g} F_u(W^{(\tau)})$$

so that

$$\mathbb{E}\langle \nabla_{W_g} F(W^{(\tau)}), \Delta_\tau \rangle = -\eta_l H \|\nabla_{W_g} F(W^{(\tau)})\|^2 + \mathbb{E}\Big\langle \nabla_{W_g} F(W^{(\tau)}), \frac{1}{|S_\tau|} \sum_{u \in S_\tau} \text{drift}_u \Big\rangle$$

Applying Young's inequality and the drift bound gives

$$\mathbb{E}\langle \nabla_{W_g} F(W^{(\tau)}), \Delta_\tau \rangle \leq -\frac{\eta_l H}{2}\|\nabla_{W_g} F(W^{(\tau)})\|^2 + \mathcal{O}(L^2 \eta_l^2 H^2 (G_g^2 + \sigma_g^2))$$

For the update norm, standard variance decomposition yields

$$\mathbb{E}\|\Delta_\tau\|^2 \leq \mathcal{O}(H^2 \eta_l^2 (G_g^2 + \sigma_g^2)) + \mathcal{O}\Big(\frac{H^2 \eta_l^2 (\sigma_g^2 + \zeta^2)}{S}\Big)$$

Combining the bounds and summing over $R$ rounds gives

$$\frac{1}{R} \sum_{\tau=0}^{R-1} \mathbb{E}\|\nabla_{W_g} F(W^{(\tau)})\|^2 \leq \frac{2(F(W^{(0)}) - F^\star)}{\eta_l H R} + \mathcal{O}(L^2 \eta_l H (G_g^2 + \sigma_g^2)) + \mathcal{O}\Big(\eta_l H \frac{\sigma_g^2 + \zeta^2}{S}\Big)$$

and hence

$$\min_{\tau \in \{0,\dots,R-1\}} \mathbb{E}\|\nabla_{W_g} F(W^{(\tau)})\|^2 \leq \frac{2(F(W^{(0)}) - F^\star)}{\eta_l H R} + \mathcal{O}(L^2 \eta_l H (G_g^2 + \sigma_g^2)) + \mathcal{O}\Big(\eta_l H \frac{\sigma_g^2 + \zeta^2}{S}\Big)$$

This completes the proof.

## C. Experimental Details

For a fair comparison, we set latent dimension to 32, batch size to 256, local epochs to 5, and communication rounds to 100, which suffices for convergence. We evaluate at $K = 10$ and tune the learning rate over $\{0.0001, 0.0005, 0.001\}$ for all baselines, while retaining other hyperparameters from the original papers. For the diffusion model, we set the number of diffusion steps to $T = 32$.

---

**Algorithm 1** Joint Training of FedDistRec

---

**ServerProcedure**:

1: Initialize global item embedding matrix $E_{\mathcal{I}}^{(1)}$ and global denoising predictor $Q_{\xi^{(g)}}^{(1)}$

2: **for** each round $r = 1, 2, \ldots, R$ **do**

3:  Randomly select client subset $B^{(r)}$ from $N$ clients with ratio $\beta$

4:  **for** client $u \in B^{(r)}$ **in parallel do**

5:   $(E_{\mathcal{I}}^u, Q_{\xi^{(g)}}^u) \leftarrow$ ClientUpdate$(u, r, E_{\mathcal{I}}^{(r)}, Q_{\xi^{(g)}}^{(r)})$

6:  **end for**

7:  $E_{\mathcal{I}}^{(r+1)}, Q_{\xi^{(g)}}^{(r+1)} \leftarrow$ GlobalAgg$(\{E_{\mathcal{I}}^u\}_{u \in B^{(r)}}, \{Q_{\xi^{(g)}}^u\}_{u \in B^{(r)}})$

8: **end for**

**ClientUpdate**$(u, r, E_{\mathcal{I}}^{(r)}, Q_{\xi^{(g)}}^{(r)})$:

1: **if** $r = 1$ **then**

2:  Initialize user-specific parameters (*e.g.,* $H_\psi, Q_{\xi^{(p)}}, S_\phi$)

3: **else**

4:  Load user-specific parameters from previous round

5: **end if**

6: Initialize $(E_{\mathcal{I}}, Q_{\xi^{(g)}})$ with $(E_{\mathcal{I}}^{(r)}, Q_{\xi^{(g)}}^{(r)})$

7: Derive user-specific semantic condition $\mathbf{c}_u$ via Eq. (8)

8: **for** $h = 1, 2, \ldots, H$ **do**

9:  Update all trainable parameters via Eq.(13)

10: **end for**

11: **Return** $E_{\mathcal{I}}, Q_{\xi^{(g)}}$

---

**Algorithm 2** Local Inference and Recommendation of FedDistRec

---

**ClientInference**$(u)$:

1: Draw $\{\mathbf{x}_T^{(s)}\}_{s=1}^{N_s}$ from $\mathcal{N}(\mathbf{0}, \mathbf{I})$

2: Derive user-specific semantic condition $\mathbf{c}_u$ via Eq. (8)

3: **for** $t = T, T-1, \ldots, 1$ **do**

4:  Compute $\{\mathbf{x}_{t-1}^{(s)}\}_{s=1}^{N_s}$ via Eq.(16)

5: **end for**

6: Calculate the prediction scores using Eq.(17) and Eq.(18)

7: **Return** $\{\bar{s}_{ui}\}$

---

## D. Additional Experiments

### D.1. Further Ablation Study Results

Figure 5 summarizes the ablation study results on HR@10, NDCG@10, Recall@10, and Precision@10. Experimental results on all datasets show that our method outperforms other ablation variants across all metrics, confirming the effectiveness of **Category Semantic Conditioning** and **Global–Personalized Denoising Parameterization**.

### D.2. Further Hyper-Parameter Analysis Results

**Regularization Coefficient** $\lambda$. We evaluate $\lambda$ over $\{0.01, 0.05, 0.1, 0.5, 1.0\}$. As illustrated in Figure 6 and 7, a small value (*e.g.,* 0.01) weakens the reconstruction objective, impairing latent representation learning. Conversely, a large $\lambda$ (*e.g.,* 1.0) biases the model toward reconstruction, compromising the capture of user preferences, highlighting the critical role of $\lambda$ in balancing reconstruction faithfulness and preference modeling.

**Number of Inference Samples** $N_s$. We analyze the effect of inference sample count, varying $N_s$ over $\{1, 3, 5, 8, 10\}$. As shown in Figures 8 and 9, a single inference sample yields a high-variance approximation, insufficient for modeling user interest uncertainty. Increasing $N_s$ boosts performance, particularly in the low-sample regime. However, the performance curve flattens as $N_s$ grows, rendering additional samples ineffective despite the linear rise in computational overhead.

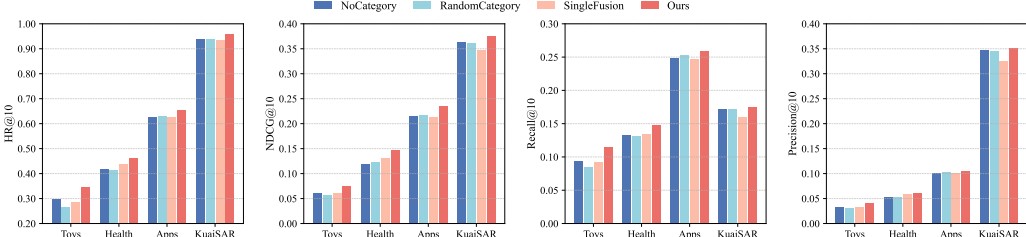

*Figure 5.* Comprehensive ablation study for category semantic conditioning and global–personalized denoising parameterization.

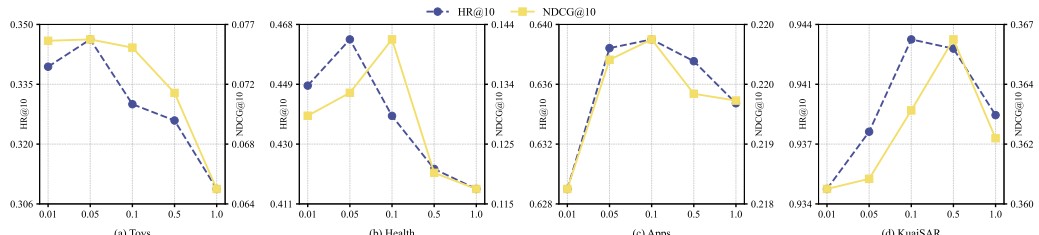

*Figure 6.* Hyperparameter analysis of regularization coefficient $\lambda$ on all datasets (HR@10, NDCG@10)

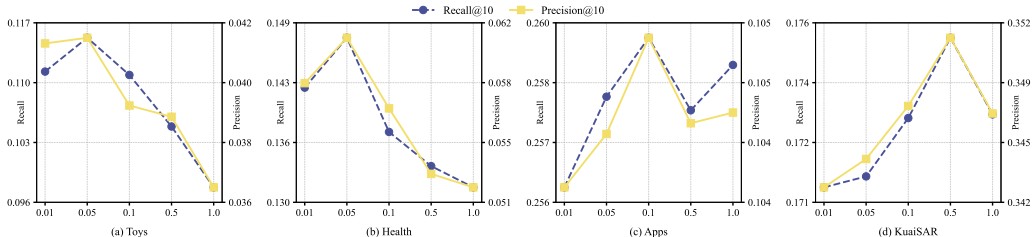

*Figure 7.* Hyperparameter analysis of regularization coefficient $\lambda$ on all datasets (Recall@10, Precision@10)

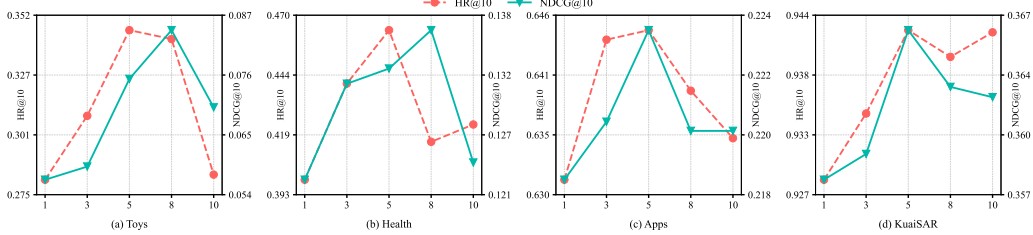

*Figure 8.* Hyperparameter analysis of number of inference samples $N_s$ on all datasets (HR@10, NDCG@10)

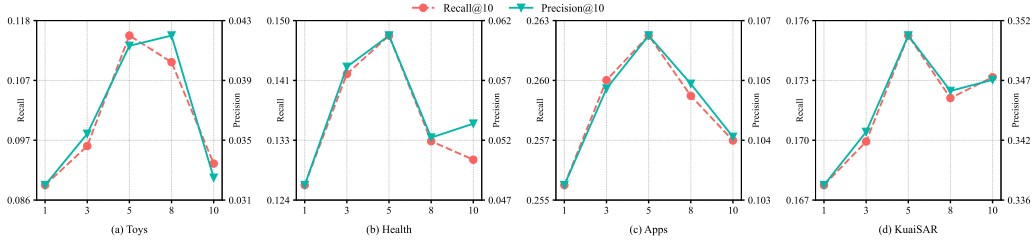

*Figure 9.* Hyperparameter analysis of number of inference samples $N_s$ on all datasets (Recall@10, Precision@10)

