# OpenReview forum: "Beyond Single Embedding: Modeling User Preferences as Distribution in Federated Recommendation"
_ICML.cc/2026/Conference — ICML 2026 regular_

### Official Review · Reviewer_PQPw · 2026-03-04

**Soundness:** 4
**Presentation:** 4
**Significance:** 4
**Originality:** 4
**Overall Recommendation:** 5
**Confidence:** 4

**Summary:**

The authors frame user embedding learning as a diffusion-based generative task using a progressive denoising process, preserving uncertainty and diversity in user representations. Experimental results demonstrate the framework’s effectiveness, and analyses of prediction score distributions and user-level ranking violation rates provide insight into its improved discrimination between positive and negative samples.

**Compliance With Llm Reviewing Policy:**

Affirmed.

**Final Justification:**

Great work.

**Key Questions For Authors:**

W1: While papers [1] and [2] also introduce diffusion models into federated recommender systems, they are not included among the diffusion-based baselines for comparison. The omission of these relevant works raises concerns about the comprehensiveness of the experimental evaluation.
W2: Clarification is needed on whether the prediction head $g_\omega$ is shared or kept private during federated aggregation.

[1] Federated Recommender System Based on Diffusion Augmentation and Guided Denoising
[2] FedDiffRec: A Module-wise Training Approach for Diffusion-Based Recommendation in Federated Learning

**Limitations:**

N.A.

**Strengths And Weaknesses:**

S1: The model incorporates category-level semantics of interacted items into the diffusion process and employs a dual-branch denoising network, combining a shared global predictor with a client-side predictor to balance personalization and generalization efficiently.
S2: The authors evaluate their method against baselines spanning three categories (box-embedding, diffusion-based, and federated methods) on four datasets of different scales and sparsity levels. The results consistently show the superiority of the proposed approach.
S3: The paper is well-structured, with a clear presentation that systematically introduces the methodology, making the proposed approach easy to follow and understand. The availability of open-sourced code supports reproducibility.

---

> ### Author Rebuttal · Authors · 2026-03-31
>
> ### Q1: [Weakness 1] Omission of some diffusion-based baselines
>
> We appreciate the reviewer’s comment regarding the omission of papers [1] and [2] as diffusion-based baselines. These papers were excluded because they use diffusion models for **data augmentation**, generating pseudo-interaction records to mitigate sparsity, whereas our approach directly **models user representation distributions via a diffusion-based generative process**, enhancing the modeling of diverse user preferences. Due to this difference in application, a direct experimental comparison is not straightforward. Nonetheless, we acknowledge the relevance of these works and will discuss their approaches and distinctions in the revised manuscript.
>
> [1] Di Y, Shi H, Wang X, et al. Federated recommender system based on diffusion augmentation and guided denoising[J]. ACM Transactions on Information Systems, 2025, 43(2): 1-36.
>
> [2] Li G, Zhang L, Rong Q, et al. FedDiffRec: A Module-wise Training Approach for Diffusion-Based Recommendation in Federated Learning[C]//ICASSP 2025-2025 IEEE International Conference on Acoustics, Speech and Signal Processing (ICASSP). IEEE, 2025: 1-5.
>
>
> ### Q2: [Weakness 2] Clarification on prediction head $g_{\omega}$
> Thank you for your comment. Since our proposed FedDistRec can be seamlessly integrated into federated recommendation methods, whether the prediction head $g_{\omega}$ is shared or private during aggregation depends on the backbone architecture. Specifically, for FedMF and FedRAP, the prediction heads are dot products, which are non-parameterized operations and do not require aggregation. In contrast, for FedNCF and PFedRec, the prediction heads are MLPs, which are shared and aggregated following the procedures in their original implementations.

---

> > ### Author Rebuttal · Reviewer_PQPw · 2026-04-01
> >
> > Thanks for your response. Great job.

---

### Official Review · Reviewer_aKnZ · 2026-03-05

**Soundness:** 4
**Presentation:** 3
**Significance:** 4
**Originality:** 3
**Overall Recommendation:** 5
**Confidence:** 5

**Summary:**

This paper introduces FedDistRec, a novel federated recommendation framework that models user representations as distributions instead of conventional fixed user embeddings, thereby achieving a more nuanced capture of user preferences in federated settings. By leveraging the distribution modeling capabilities of diffusion models, FedDistRec demonstrates superior performance across multiple datasets.

**Compliance With Llm Reviewing Policy:**

Affirmed.

**Final Justification:**

The author's response addressed my concerns. I have no further questions.

**Key Questions For Authors:**

Please address the concerns in the Weaknesses.

**Strengths And Weaknesses:**

Strengths:
1. The motivation of the study is clearly articulated. This paper highlights the limitations of representing users with a single embedding and introduces the idea of modeling user representations as distributions in the federated recommendation setting. This perspective is conceptually meaningful and offers a potentially valuable extension to existing approaches.
2. The empirical validation is comprehensive. In particular, comparisons with single embedding-based baselines and the analysis of distribution-induced ranking behavior jointly support the central claim of modeling user representations as distributions in the federated recommendation setting, offering a solid empirical foundation for the proposed approach.
3. The proposed mechanism exhibits practical applicability. It can be readily integrated into existing federated recommendation frameworks with minimal modification.

Weaknesses
1. The paper references several non-single-user-embedding methods but lacks an in-depth comparative discussion with these approaches, which provides limited insight into the relative advantages and contributions of the proposed method.
2. Some key aspects require clarification. Specifically, a clear definition of "interaction diversity" should be provided in Section 5.5.

---

> ### Author Rebuttal · Authors · 2026-03-31
>
> ### Q1: [Weakness 1] In-depth comparison with non-single-user-embedding methods
>
> We thank you for your valuable feedback. In this response, we discuss representative non-single-user-embedding methods in our experiments: HCUR, LCD-UC, and GCDR. HCUR and LCD-UC represent users as hypercuboids, increasing preference modeling capacity compared to a single vector, but still constrained to a fixed region in embedding space. GCDR maintains multiple latent distributions per client, but in federated settings, it mainly captures shared patterns across users while neglecting individual user heterogeneity.
>
> In contrast, our FedDistRec learns user representation distributions via a diffusion-based generative model, better capturing diverse preferences (**superior to HCUR and LCD-UC**). Its Global–Personalized Denoising Parameterization integrates global patterns with client-specific characteristics, enabling expressive and personalized modeling (**superior to GCDR**). These advantages are further validated by tok-k recommendation and distribution-induced ranking experiments.
>
> ### Q2: [Weakness 2] Explanation of "interaction diversity" in Section 5.5
>
> Thank you for your valuable comment. In Section 5.5, interaction diversity refers to the number of distinct item categories that a user has interacted with in their training data.

---

> > ### Author Rebuttal · Reviewer_aKnZ · 2026-04-01
> >
> > The author's response addressed my concerns. I have no further questions.

---

### Official Review · Reviewer_QU84 · 2026-03-12

**Soundness:** 4
**Presentation:** 3
**Significance:** 3
**Originality:** 4
**Overall Recommendation:** 5
**Confidence:** 5

**Summary:**

This paper studies user personalization modeling in federated recommender systems. Existing methods typically achieve personalization by learning user-specific parameters or leveraging information from similar users. The paper proposes to model user representations as distributions using a diffusion model based on users’ interaction histories. During recommendation, multiple user representations are sampled from the learned distribution and used jointly for prediction. Experimental results show that the proposed approach achieves competitive recommendation performance.

**Compliance With Llm Reviewing Policy:**

Affirmed.

**Key Questions For Authors:**

1. How does modeling user representations as distributions differ from existing federated recommendation personalization techniques, and why are current methods insufficient for capturing user-specific preferences?

2. What is the computational overhead of the proposed method, and how does it compare to existing approaches?

3. The denoising network relies on item category information as input, does the total number of categories have a significant impact on model performance?

**Limitations:**

Please address the questions listed above.

**Strengths And Weaknesses:**

**Strengths:**

S1: The paper presents a novel perspective on personalized modeling in federated recommender systems by representing users as distributions. This formulation differs from existing approaches that rely on user-specific parameters or similarity-based information sharing, and provides a more flexible way to capture variations in user preferences.

S2: The proposed framework adopts an intuitive modeling strategy by learning user representation distributions directly from users’ historical interactions. This design aligns with standard recommendation settings and does not require additional auxiliary information, keeping the framework relatively simple and broadly applicable.

S3: The paper is clearly written and well organized, making it easy to follow.

**Weaknesses:**

W1: The paper lacks an in-depth discussion of certain key questions, such as how modeling user representations as distributions differs from existing federated recommendation personalization techniques and why current methods may be insufficient for capturing user-specific preferences.

W2: The framework relies on a diffusion-based generative model, which could introduce non-negligible computational overhead.

---

> ### Author Rebuttal · Authors · 2026-03-31
>
> ### Q1: [Questions 1 & Weaknesses 1] How distributional user modeling differs from existing methods
> We thank the reviewer for the comment. Existing work in federated recommendation has primarily focused on **parameter-level personalization**. For example, some methods personalize **item embeddings** for each user [1,2], while others maintain **user-specific scoring functions** [3]. Despite these efforts, each user is still represented by a **single embedding vector**, which may be insufficient to capture the variability and uncertainty in user preferences, particularly under sparse interactions.
>
> In contrast, our approach **models user representations as distributions** rather than single vectors. During inference, multiple user embeddings can be sampled from the learned distribution and aggregated to generate recommendations, providing a richer and more flexible representation of user behavior. Moreover, parameter-level personalization can be combined with our distributional user representations; as validated in our experiments, this integration further improves the modeling of user-specific preferences and enhances recommendation performance.
>
> [1] Li Z, Long G, Zhou T. Federated Recommendation with Additive Personalization[C]//The Twelfth International Conference on Learning Representations. 2024.
>
> [2] Zhang C, Long G, Zhou T, et al. Gpfedrec: Graph-guided personalization for federated recommendation[C]//Proceedings of the 30th ACM SIGKDD conference on knowledge discovery and data mining. 2024: 4131-4142.
>
> [3] Zhang C, Long G, Zhou T, et al. Dual personalization on federated recommendation[C]//Proceedings of the Thirty-Second International Joint Conference on Artificial Intelligence. 2023: 4558-4566.
>
> ### Q2: [Questions 2 & Weaknesses 2] Computational Overhead Comparison
>
> We thank the reviewer for raising the question regarding computational overhead. We use PFedRec as the backbone and report model parameters, average epoch training time in milliseconds, and HR@10 on the Toys dataset (Table below). Incorporating FedDistRec increases both the number of parameters and the epoch training time compared with PFedRec, but yields substantial improvements in HR@10 (12.8%). Compared with other baselines, FedDistRec achieves comparable overall computational cost and is even more efficient in some cases, while consistently delivering higher HR@10, demonstrating a favorable trade-off between computational cost and recommendation performance.
>
>
> | Methods          | Parameter Count | Time (ms)       | HR@10            |
> |:----------------:|:---------------:|:--------------:|:----------------:|
> | HCUR             | 31,680           | 756.25         | 29.49            |
> | LCD-UC           | 38,016           | 692.01         | 31.54            |
> | DDRM             | 36,864           | 703.41         | 30.63            |
> | GCDR             | 35,360           | 759.82         | 29.78            |
> | PFedRec          | 28,097           | 682.02         | 30.66            |
> | PFedRec + FedDistRec | 33,761 (+20.1%) | 710.20 (+4.1%) | 34.59 (+12.8%)   |
>
>
>
> ### Q3: [Questions 3] Effect of category number on performance
> We thank the reviewer for this question. In our experiments, the datasets encompass a diverse range of category scales. The results consistently demonstrate that incorporating item category information yields notable performance gains, remaining robust across datasets with varying category granularities.

---

> > ### Author Rebuttal · Reviewer_QU84 · 2026-04-01
> >
> > Thank you for the response, I maintain my response.

---

### Official Review · Reviewer_r8Ti · 2026-03-13

**Soundness:** 2
**Presentation:** 3
**Significance:** 2
**Originality:** 3
**Overall Recommendation:** 4
**Confidence:** 3

**Summary:**

This paper proposes FedDistRec, a federated recommendation framework that represents each user as a distribution over the item-embedding space rather than a single point embedding. The core motivation is that in federated settings, each client observes only sparse, fragmentary interaction data, making single-point user embeddings unable to capture the inherent ambiguity and diversity of user preferences. Experiments on four datasets show consistent improvements over diffusion-, box-embedding-, and federated baselines.

**Compliance With Llm Reviewing Policy:**

Affirmed.

**Final Justification:**

My concerns are all addressed and I maintain my positive score.

**Key Questions For Authors:**

See weakness.

**Limitations:**

yes

**Strengths And Weaknesses:**

**Strength**
- The paper treat federated user modeling as distributional inference via diffusion, directly addressing preference ambiguity that arises when each client observes only sparse, fragmentary interactions.
-  Splitting the denoising function into a globally shared predictor and a locally private predictor directly mirrors the generalization/personalization trade-off inherent in federated settings. This decomposition is cleanly integrated into the diffusion framework rather than being an ad-hoc addition, and the adaptive fusion module provides a learnable mechanism for balancing the two views.
- Broad comparison against three families of baselines (box embeddings, diffusion recommenders, classical FRS) with plug-and-play integration into several federated backbones is presented.


**Weakness**
- The construction of the clean user representation $x_0$ is under-specified. The paper states that $x_0$ is "derived from the user's interacted items" using the shared item embedding matrix $E_I$ (Section 4.1), but does not specify the aggregation mechanism.
- The paper filters users with fewer than 10 interactions (Amazon) or 50 interactions (KuaiSAR), yet the core motivation emphasizes sparse interactions and preference ambiguity. Evaluating performance on genuinely sparse users would more directly validate the claimed benefits of distributional representations. The current filtering undermines the motivating argument.
-  While the paper cites PFedRec and FedRAP, the design of splitting parameters into global and local components has been explored extensively in PFL (e.g., FedPer, FedRoD, and other decomposition-based methods). A more thorough discussion of how the proposed dual-view denoising relates to and differs from these approaches would better position the contribution.

---

> ### Author Rebuttal · Authors · 2026-03-31
>
> ### Q1: [Weaknesses 1] Clarification on clean user representation $x_0$ construction
> Thank you for your feedback. We would like to kindly clarify a potential ambiguity in the description of $x_0$ construction. In our framework, $x_0$ is **not constructed by aggregating all interacted items**. Instead, during local training, we iterate over each interacted item $i$ $\in$ $\mathcal{R}\_u$  and construct $x_0$=$E_{\mathcal{I}}(i)$, i.e., by retrieving the embedding vector of item $i$ from the item embedding table $E_{\mathcal{I}}$. The $x_0$ is then used for subsequent diffusion model training to characterize the distribution of user representations. We will revise the manuscript to make this point explicit and avoid any ambiguity.
>
>
> ### Q2: [Weaknesses 2] Clarification on dataset filtering strategy
> We thank the reviewer for this insightful comment. We adopt k-core filtering (k=10 for Amazon and k=50 for KuaiSAR) following common practice in prior work [1,2,3,4], mainly to ensure that each user provides sufficient signal for training and evaluation; under our 7:1:2 train/validation/test split, users with fewer than 10 interactions would not admit a meaningful partition, making the learning process ill-defined. Importantly, this preprocessing does not alter the fundamental sparsity of the recommendation task. Sparsity here refers to the fact that each user interacts with only a very small fraction of the entire item space. For example, KuaiSAR contains 6,882 items, and even after filtering, a user with 50 interactions still covers less than 1% of all items, indicating that the data remains highly sparse in a practical sense. Therefore, the evaluation setting continues to reflect the core challenge of limited user feedback, and the observed performance of FedDistRec provides evidence that the proposed distributional representations are effective under such intrinsically sparse conditions.
>
>
> ### Q3: [Weaknesses 3] Difference between the proposed global–personalized denoising parameterization and existing PFL methods
> We thank the reviewer for this valuable comment. Prior decomposition-based PFL methods can be broadly categorized into two types. The first type, exemplified by FedPer [5], decomposes the model into shared and private components, effectively partitioning the model’s functionality into global and local parts. The second type, such as FedRoD [6], duplicates a functional component into both global and local versions; our dual-view denoising belongs to this category.
>
> In the second type, FedRoD maintains separate global and local prediction heads and optimizes them with two independent losses. In contrast, our approach places both shared and private predictors within the diffusion denoising network, fuses their predictions, and computes a single reconstruction loss. This design encourages the model to leverage global patterns and individual-specific features jointly, providing a unified training signal that leads to more coherent user representation learning and mitigates potential conflicts between shared and private components. We will incorporate this discussion into the manuscript to clearly position our contribution.
>
> [1] Xia L, Huang C, Xu Y, et al. Hypergraph contrastive collaborative filtering[C]//Proceedings of the 45th International ACM SIGIR conference on research and development in information retrieval. 2022: 70-79.
>
> [2] Li Z, Long G, Zhou T. Federated Recommendation with Additive Personalization[C]//The Twelfth International Conference on Learning Representations. 2024.
>
> [3] He X, Liao L, Zhang H, et al. Neural collaborative filtering[C]//Proceedings of the 26th international conference on world wide web. 2017: 173-182.
>
> [4] Perifanis V, Efraimidis P S. Federated neural collaborative filtering[J]. Knowledge-Based Systems, 2022, 242: 108441.
>
> [5] Arivazhagan M G, Aggarwal V, Singh A K, et al. Federated learning with personalization layers[J]. arXiv preprint arXiv:1912.00818, 2019.
>
> [6] Chen H Y, Chao W L. On Bridging Generic and Personalized Federated Learning for Image Classification[C]//International Conference on Learning Representations. 2022.

---

> > ### Author Rebuttal · Reviewer_r8Ti · 2026-04-02
> >
> > Thank you for the response, I maintain my positive score.

---

### Decision · Program_Chairs · 2026-04-30

**Decision:**

Accept (regular)

**Comment:**

This paper introduces a distributional approach to personalized federated recommendation, effectively addressing the information bottleneck of single-vector representations in complex, non-IID environments. While reviewers initially raised valid questions regarding computational overhead and category sensitivity, the authors provided robust empirical evidence during the rebuttal to demonstrate that the significant performance gains justify the manageable trade-offs. Successfully adapting distributional preference modeling to the strict privacy and communication constraints of federated learning is non-trivial. The authors have delivered a technically sound framework. Overall, this is an advancement for the federated recommendation community. I recommend acceptance.